# Intraoperative intravenous low-dose esketamine improves quality of early recovery after laparoscopic radical resection of colorectal cancer: A prospective, randomized controlled trial

**Ying Xu<sup>☯</sup>, Long He<sup>☯</sup>, Shaoxuan Liu, Chaofan Zhang, Yanqiu Ai<sup>ⓘ</sup>***

Department of Anesthesiology, Pain and Perioperative Medicine, The First Affiliated Hospital of Zhengzhou University, Zhengzhou, Henan, China

☯ These authors contributed equally to this work.

* aiyanqiu82@163.com

## Abstract

### Background

Esketamine has higher potency, stronger receptor affinity, a stronger analgesic effect, a higher in vivo clearance rate, and a lower incidence of adverse reactions when compared to ketamine. However, there have been few ketamine studies to assess patient-centered, overall recovery outcomes from the perspective of patients with colorectal cancer.

### Methods

This was a prospective, randomized controlled trial. Ninety-two patients undergoing laparoscopic radical resection of colorectal cancer were randomly assigned to either the esketamine (K group) or non-eskatamine (C group) group. After anesthesia induction, a loading dose of 0.25 mg/kg was administered, followed by continuous infusion at a rate of 0.12 mg.kg$^{-1}$.h$^{-1}$ until closure of surgical incisions in the K group. In the C group, an equivalent volume of normal saline was infused. The primary outcome was quality of recovery at 24 h after surgery, as measured by the Quality of Recovery-15 (QoR-15) scale. The QoR-15 was evaluated at three timepoints: before (T$_{before}$), 24 h (T$_{24h}$) and 72 h (T$_{72h}$) after surgery.

### Main results

A total of 88 patients completed this study. The total QoR-15 scores in K group (n = 45) were higher than in the C group (n = 43) at 24 h: 112.33 ± 8.79 vs. 103.93 ± 9.03 (P = 0.000) and at 72 h: 118.73 ± 7.82 vs. 114.79 ± 7.98 (P = 0.022). However, the differences between the two groups only had clinical significance at 24 h after surgery. Among the five dimensions of the QoR-15, physical comfort (P = 0.003), emotional state (P = 0.000), and physical independence (P = 0.000) were significantly higher at 24 h in the K group, and physical comfort (P = 0.048) was higher at 72 h in the K group.

**Data Availability Statement:** The original contributions presented in our study are included in the paper and its Supporting Information files.

Further inquiries can be directed to the corresponding author.

**Funding:** This work was supported by the National Natural Science Foundation of China (grant number: 82001189) to Long He. and the Henan Science and Technology project (grant number: 201911225) to Yanqiu Ai. The funders had no role in study design, data collection and analysis, decision to publish, or preparation of the manuscript.

**Competing interests:** The authors have declared that no competing interests exist.

## Conclusions

This study found that intraoperative intravenous low-dose esketamine could improve the early postoperative quality of recovery in patients undergoing laparoscopic radical resection of colorectal cancer from the perspective of patients.

## 1. Introduction

Colorectal cancer (CRC) is a common clinical malignant tumor of the digestive system, with an increasing incidence in recent years [1]. CRC is currently primarily treated using comprehensive methods based on surgical resection [2]. Laparoscopic radical surgery is less invasive and has fewer postoperative complications than traditional open surgery, and it has become a common surgical method in clinical practice [3]. It dose, to some extent, hasten postoperative recovery, which is consistent with the concept of enhanced recovery after surgery (ERAS) [4]. However, pain after laparoscopic surgery is a common issue that may interfere with discharge and recovery [5]. As anesthesiologists, we are responsible for optimizing patients' postoperative experience, minimizing the incidence of postoperative adverse reactions and the recovery time of daily activities, and striving to improve the quality of postoperative recovery [6].

The choice of perioperative anesthetic drugs and techniques has a significant impact on the patient's postoperative recovery [7]. Esketamine is the dextroisomer of ketamine and an N-methyl-D-aspartate (NMDA) receptor antagonist. Its potency is twice that of ketamine, resulting in stronger sedative and analgesic effects [8, 9]. Because of the dose-dependent side effects of ketamine, esketamine has a lower likelihood of adverse reactions (such as psychedelic effects) [10]. Furthermore, esketamine has a high clearance rate and rapid metabolism in the human body, which improves the controllability of anesthesia and allows patients to wake up faster and safer [11]. Recent ketamine research in patients with colorectal cancer has primarily focused on traditional recovery indicators such as fatigue syndrome, emotional responses, and inflammatory factors, pain, and tumor growth [12–15]. There is a scarcity of data to assess overall recovery outcomes from the perspective of patients with colorectal cancer. The 15-item Quality of Recovery scale (QoR-15) is a patient-centered outcome measure used to assess patients' health status following surgery and anesthesia. Furthermore, the QoR-15 is dependable, valid and has high clinical acceptability [16, 17].

Therefore, the objective of our study was to assess the effects of intraoperative intravenous low-dose esketamine on the quality of early recovery in patients undergoing laparoscopic radical resection of CRC using the QoR-15 scale.

## 2. Materials and methods

### 2.1 Study design and participants

This prospective, randomized controlled trial has been approved by the Ethics Committee of the First Affiliated Hospital of Zhengzhou University (2021-KY-1097-003). The trial was pre-registered at the Chinese Clinical Trial Registry (ChiCTR2200056823) on February 18, 2022. Written informed consent was obtained from all patients.

Ninety-two patients aged from 18 to 65 years, with a BMI of 18–30 kg/m$^2$ and American Society of Anesthesiologists (ASA) physical status I-II, who were scheduled for laparoscopic radical resection of CRC at the First Affiliated Hospital of Zhengzhou University were enrolled in this study between 25 February 2022 and 30 August 2022. The exclusion criteria were as

follows: refusal to participate in the study; esketamine contraindications such as glaucoma, large vessel aneurysm, etc.; preoperative use of sedative or analgesic drugs; severe cardiopulmonary, liver, and kidney dysfunction; cognitive impairment or a history of psychiatric or neurological disorders.

Using a random number table method, patients were randomly assigned in a 1:1 ratio to esketamine group (K group) and control group (C group). Each patient's group arrangement was stored in a sealed envelope and opened by the anesthesia nurse who was not involved in the study when the patient entered the operation room. Anesthesiologists were unaware of the patient grouping and administered anesthesia to patients. Patients, surgeons, and postoperative follow-up assessors (the same anesthesiologists) were blinded to the group assignment. Esketamine, 50 mg, was diluted to a total of 50 ml with normal saline solution. Both esketamine and normal saline were drawn into identical 50-ml syringes by the same anesthesia nurse.

All patients fasted for 8 hours and were denied water for 4 hours before surgery and peripheral venous access was established before the patients entered the operation room. After patients entered the operation room, peripheral venous access was opened and the electrocardiogram (ECG), non-invasive blood pressure (NBP), heart rate (HR), pulse oxygen saturation (SpO$_2$), and the bispectral index (BIS) were all monitored. The radial artery and right internal jugular vein were punctured and catheterized under local anesthesia. After 3–5 minutes of preoxygenation, all patients were induced for general anesthesia by intravenous injection of 0.2–0.3 mg/kg etomidate, 0.5–1.0 µg/kg sufentanil, 0.15–0.3 mg/kg cisatracurium, and maintained with 4–12 mg.kg$^{-1}$.h$^{-1}$propofol and 0.05–2 ug.kg$^{-1}$.min$^{-1}$ remifentanil after intubation. The BIS value of the subjects was kept between 40 and 60 by adjusting the propofol infusion rate during surgery. The oxygen and air inhalation flow were set as 1 L/min respectively, V$_T$ 6–8 ml/kg, RR 12–16 times/min, I: E = 1:2 and P$_{ET}$CO$_2$ 35–45 mmHg (1mmHg = 0.133kPa). To maintain neuromuscular blockade during surgery, an intermittent intravenous injection of cisatracurium was used. When necessary, vasoactive drugs were used to regulate hemodynamic stability. After anesthesia induction, a loading dose of 0.25 mg/kg of esketamine was administered, followed by continuous infusion at a rate of 0.12 mg.kg$^{-1}$.h$^{-1}$ until closure of surgical incisions in the K group. In the C group, an equivalent volume of normal saline was infused.

Palonosetron (0.25mg) was injected intravenously to prevent nausea and vomiting, and sufentanil (0.1–0.2 µg/kg) was used for analgesia about 30 minutes before the end of the surgery. All anesthesia maintenance drugs were stopped when the skin was sutured. Following surgery, all patients in both groups received a patient-controlled intravenous analgesia pump (PCIA, 200 ml) containing oxycodone 0.6 mg/kg and palonosetron 0.25 mg for 48 hours. The background infusion rate was 3 ml/h, the single additional dose was 4 ml, the locking time was 15 minutes, and no initial infusion dose was administered. The visual analogue scale (VAS) was used to assess pain intensity; a score of 0 was defined as no pain, and a score of 10 was defined as worst imaginable pain. After the operation, all patients were transferred to the postanesthesia care unit (PACU). When the VAS score was ≥ 4, oxycodone 0.05–0.1 mg/kg was injected intravenously for rescue analgesia during the stay in PACU. Patients relieved pain after returning to the ward by increasing the number of analgesic pump compressions and additional analgesic drugs.

We excluded patients from the study if they were changed to laparotomy, lost to follow-up, the bleeding volume was greater than 500ml, a large amount of subcutaneous emphysema during the operation, or hypercapnia after operation could not be corrected in a short period of time, the surgical procedure was Miles, and severe postoperative complications impacted the quality of postoperative recovery.

The primary outcome of this study was the quality of recovery at 24 h after surgery, as measured by the QoR-15 scale. The QoR-15 measures 15 items from five domains of recovery to

assess a patient's overall recovery: physical comfort (5 items), emotional state (4 items), psychological support (2 items), physical independence (2 items), and pain (2 items). Each item is scored from 0 to 10, with 0 indicating no existence and 10 indicating continuous or strong existence; however, the negative indicators were scored in reverse. The overall QoR-15 score ranges from 0 to 150, with higher scores indicating better quality of postoperative recovery. The QoR-15 was assessed at the following three timepoints: before ($T_{before}$), 24 h ($T_{24h}$) and 72 h ($T_{72h}$) after surgery.

The secondary outcomes were as follows: the duration of surgery, the consumption of remifentanil, the dosage of cisatracurium, the use of vasoactive drugs, blood loss, rescue analgesia during PACU, respiratory recovery time (the time from anesthetic withdrawal to spontaneous breathing), and extubation time (the time from anesthetic withdrawal to extubation); MAP (mean arterial pressure) and HR at before induction ($T_0$), after carbon dioxide pneumoperitoneum establishment ($T_1$), after carbon dioxide pneumoperitoneum closure ($T_2$), and at the end of surgery ($T_3$); VAS pain scores at rest (VAS-R) and on coughing (VAS-M) at 4 h, 8 h, 12 h, 24 h and 48 h postoperatively; times of effective PCIA press at 24 h and 48 h after operation, the consumption of oxycodone, the use of additional analgesic drugs, hospital stay duration, intestinal function recovery time (the duration of time from extubation to the first exhaust time), time of first eating or drinking (the duration of time from extubation to the first drinking or eating time), ambulation time (the duration of time from extubation to the first bed time standing or walking), and adverse events such as nausea, vomiting, dizziness, hallucination and rash.

## 2.2 Sample size calculation

The primary outcome of this study was the quality of recovery score (QoR-15) at 24 h postoperatively. PASS 15.0 was used to calculate the estimated sample size based on the primary outcome. The minimal clinically important difference (MCID) in the QoR-15 scale was 8.0 [18]. According to previous research and preliminary findings, the total QoR-15 score at 24 h postoperatively in the esketamine group was 114 ± 16, compared to 102 ± 22 in the control group. A sample size of 42 subjects in each group was calculated using the preliminary study data, assuming Type I error ($\alpha$)=0.05 and Type II error ($\beta$)=0.2. To allow for a dropout rate of 10%, we finally enrolled 46 subjects in each group in this study.

## 2.3 Statistical analysis

The SPSS21.0 software (IBM) was used for statistical analysis. The Shapiro-Wilk test was used to test the data's normality. Data with a normal distribution were presented as mean and standard deviation (SD), with group comparisons analyzed using the Student's t-test. Data with normal distribution at multiple time points were analyzed using repeated measures analysis of variance (ANOVA). Data with non-normal distribution were expressed as the median (quartiles). The Mann-Whitney U test was used for intergroup comparisons. Data with non-normal distribution at multiple time points were analyzed using the Generalized Estimating Equations (GEE). The categorical data were shown as numbers (%) and were analyzed using the chi-square test or Fisher's exact test as appropriate. Adjustment for multiple testing was done using Bonferroni or LSD. A two-sided test P-value less than 0.05 was considered statistically significant.

## 3. Results

In total, 92 eligible patients were enrolled and randomly assigned to one of two groups: saline group (C group, n = 46) or esketamine group (K group, n = 46). Three patients in the C group

were excluded from the final analysis (converted to laparotomy, n = 2; and bleeding volume > 500ml, n = 1), and one patient in the K group was also excluded due to intraoperative conversion to laparotomy. Eventually, 88 patients were included in the completed study, including 45 in the K group and 43 in the C group (Fig 1). The baseline characteristics of the subjects in our study were not significantly different between the two groups (Table 1).

## 3.1 Primary outcome

The primary outcome, which is the QoR-15 scores at 24 h after surgery, is shown in Table 2. The K group had a higher total QoR-15 scores of 112.33 ± 8.79 vs. 103.93 ± 9.03 in the C group ($P = 0.000$). Among the five dimensions of the QoR-15, physical comfort ($P = 0.003$), emotional state ($P = 0.000$), and physical independence ($P = 0.000$) were significantly higher in the K group compared to the C group at 24 h after surgery (Table 2).

## 3.2 Secondary outcomes

As shown in Table 2, there was no statistically significant difference in the preoperative QoR-15 score between the two groups ($P = 0.805$). Similar to the primary outcome, the total QoR-15 score at 72 h after surgery was higher in the K group compared with the C group: 118.73 ± 7.82 vs. 114.79 ± 7.98 ($P = 0.022$). Compared with the preoperative QoR-15 score, it decreased significantly within 24 h and 72 h after surgery.

Compared with the C group, the VAS-R scores in the K group were lower at 4 h, 8 h, 12 h, 24 h and 48 h after surgery ($P = 0.043, 0.002, 0.011, 0.000, 0.047$, respectively). Compared with the C group, the VAS-M scores in the K group were lower at 8 h and 12 h after surgery ($P = 0.000, 0.002$, respectively). However, the VAS-M scores at 4 h, 24 h and 48 h after surgery were statistically similar between the two groups ($P = 0.852, 0.111, 0.602$, respectively, Fig 2).

The intraoperative data are shown in Table 3. The K group had fewer patients using vasoactive drugs than the C group ($P = 0.003$); the respiratory recovery time ($P = 0.042$) was shorter in the K group; there were no significant differences in the duration of surgery ($P = 0.732$), remifentanil dosage ($P = 0.097$), cisatracurium dosage ($P = 0.555$), infusion amount ($P = 0.061$), urine amount ($P = 0.064$), blood loss ($P = 0.211$), extubation time ($P = 0.218$), or emergence agitation ($P = 0.680$) between two groups. Compared with the values in the C group, MAP was higher at $T_2$ in the K group ($P = 0.035$, Fig 3). The postoperative data are shown in Table 4. When compared to the C group, the K group had lower use of rescue analgesia in PACU ($P = 0.045$) and additional analgesic drugs in the ward ($P = 0.045$), fewer times of effective PCIA press at 24 h ($P = 0.000$) and between 24 h and 48 h postoperatively ($P = 0.000$), and less consumption of oxycodone at 24 h ($P = 0.002$); The intestinal function recovery time ($P = 0.000$), and ambulation time ($P = 0.000$) were shorter in the K group. There were no significant differences in oxycodone consumption between 24 h and 48 h postoperatively ($P = 0.176$), the time of first eating or drinking ($P = 0.254$), or the length of postoperative hospital stay ($P = 0.279$) between the two groups. The reported adverse events are shown in Table 5.

## 4. Discussion

As anesthesia safety and surgical techniques have improved, the assessment of quality of postoperative recovery has become the primary endpoint outcome in research. In this prospective, RCT, we discovered that intravenous low-dose esketamine combined with general anesthesia improved the early postoperative quality of recovery from the patients' perspective in patients undergoing laparoscopic radical resection of CRC using the QoR-15 score as a patient-

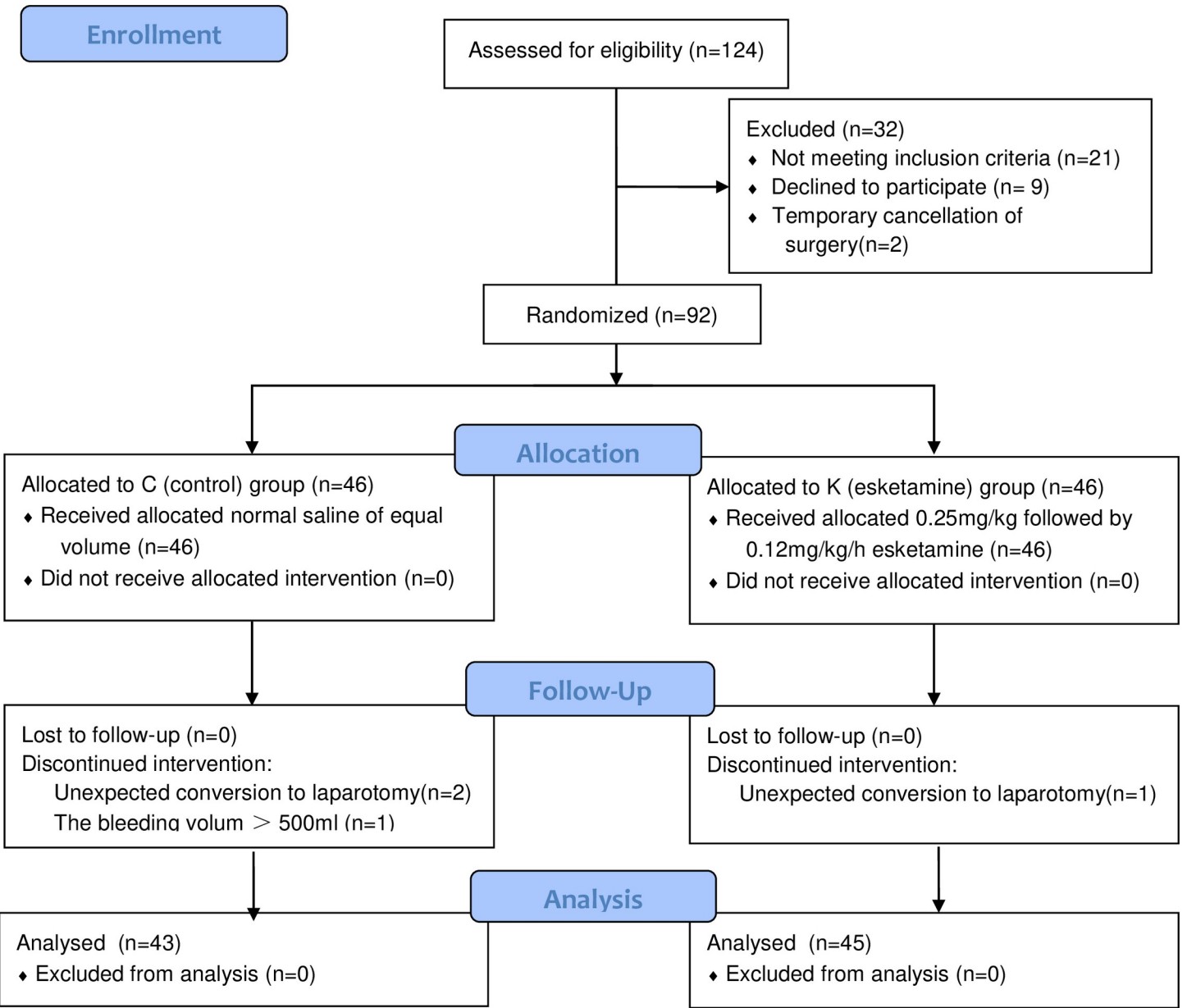

**Fig 1. CONSORT 2010 flow diagram.**

centered primary outcome. This finding has significant implications for the selection of anesthesia methods and drugs in patients undergoing laparoscopic radical resection of CRC.

Esketamine, the S(+)-isomer of ketamine, is a chiral cyclohexanone with stronger sedative and analgesic effects and is also a dissociative anesthetic; It mainly acts on NMDA receptors by binding to the phencyhexidine site, non-competitively inhibiting the activation of the receptor by glutamate, thereby weakening neuronal activity and producing sedative, analgesic and anesthetic effects [19]. Esketamine has twice the potency of ketamine, and the esketamine dose in this study was chosen based on the reference [20] and potential adverse reactions related to esketamine.

**Table 1. Patient characteristics in the two groups.**

|  | C group (n = 43) | K group (n = 45) | $t/\chi^2/Z$-value | *P*-value |
|---|---|---|---|---|
| Age (y) | 57.0 (50.0,59.0) | 56.0 (48.5,59.0) | -0.226 | 0.821 |
| Sex, n (%) |  |  | 0.060 | 0.807 |
| Male | 18 (41.9) | 20 (44.4) |  |  |
| Female | 25 (58.1) | 25 (55.6) |  |  |
| Weight (kg) | 64.7 ± 7.9 | 65.5 ± 8.6 | -0.474 | 0.637 |
| BMI (kg/m$^2$) | 24.5 ± 2.2 | 24.0 ± 2.3 | 1.105 | 0.272 |
| ASA class, n (%) |  |  | 0.839 | 0.360 |
| I | 15 (34.9) | 20 (44.4) |  |  |
| II | 28 (65.1) | 25 (55.6) |  |  |
| Type of disease, n (%) |  |  | 0.369 | 0.544 |
| Colon cancer | 18 (41.9) | 16 (35.6) |  |  |
| Rectal cancer | 25 (58.1) | 29 (64.4) |  |  |

Data are presented as mean ± SDs, number of patients (%), or median (interquartile range) as appropriated. C group = control group; K group = esketamine group; BMI = body mass index, ASA = American Society of Anesthesiologists.

**Table 2. QoR-15 scores between two groups at different time points.**

|  | C group (n = 43) | K group (n = 45) | Mean Difference (K - C) | 95%CI for Difference Lower Upper | *P*-value |
|---|---|---|---|---|---|
| Total QoR-15 |  |  |  |  |  |
| $T_{before}$ | 132.4 ± 5.4 | 132.1 ± 5.4 | - 0.285 | - 2.580 2.009 | 0.805 |
| $T_{24h}$ | 103.9 ± 9.0$^*$ | 112.3 ± 8.8$^*$ | 8.403 | 4.626 12.180 | 0.000 |
| $T_{72h}$ | 114.8 ± 8.0$^*$ | 118.7 ± 7.8$^*$ | 3.943 | 0.593 7.292 | 0.022 |
| QoR-15 domains |  |  |  |  |  |
| Physical comfort |  |  |  |  |  |
| $T_{before}$ | 42.3 ± 3.2 | 42.6 ± 3.0 | 0.320 | - 0.978 1.617 | 0.625 |
| $T_{24h}$ | 36.4 ± 5.1$^*$ | 39.8 ± 5.1$^*$ | 3.360 | 1.202 5.518 | 0.003 |
| $T_{72h}$ | 40.3 ± 5.0 | 42.3 ± 4.6 | 2.055 | 0.022 4.088 | 0.048 |
| Emotional state |  |  |  |  |  |
| $T_{before}$ | 34.0 (33.0,35.0) | 34.0 (33.0,35.0) | - 0.241 | - 0.985 0.502 | 0.524 |
| $T_{24h}$ | 32.0 (28.0,35.0) $^*$ | 36.0 (32.0,38.0) | 3.424 | 1.606 5.242 | 0.000 |
| $T_{72h}$ | 33.0 (31.0,36.0) | 34.0 (32.5,35.0) | 0.480 | - 0.672 1.631 | 0.414 |
| Psychological support |  |  |  |  |  |
| $T_{before}$ | 19.0 (18.0,20.0) | 18.0 (17.0,20.0) | - 0.349 | - 0.841 0.142 | 0.163 |
| $T_{24h}$ | 19.0 (18.0,20.0) | 19.0 (18.0,20.0) | - 0.265 | - 0.797 0.268 | 0.330 |
| $T_{72h}$ | 20.0 (17.0,20.0) | 20.0 (18.0,20.0) | 0.237 | - 0.479 0.953 | 0.517 |
| Physical independence |  |  |  |  |  |
| $T_{before}$ | 18.0 (17.0,20.0) | 19.0 (17.0,20.0) | 0.345 | - 0.309 1.000 | 0.301 |
| $T_{24h}$ | 2.0 (2.0,3.0) $^*$ | 4.0 (3.0,4.5) $^*$ | 1.409 | 0.945 1.873 | 0.000 |
| $T_{72h}$ | 6.0 (5.0,6.0) $^*$ | 6.0 (5.0,7.0) $^*$ | 0.510 | - 0.059 1.079 | 0.079 |
| Pain |  |  |  |  |  |
| $T_{before}$ | 20.0 (18.0,20.0) | 20.0 (17.0,20.0) | - 0.360 | - 1.077 0.357 | 0.325 |
| $T_{24h}$ | 15.0 (14.0,15.0) $^*$ | 15.0 (14.5,16.0) $^*$ | 0.475 | - 0.170 1.120 | 0.149 |
| $T_{72h}$ | 17.0 (16.0,18.0) $^*$ | 18.0 (17.0,20.0) $^*$ | 0.662 | - 0.210 1.533 | 0.137 |

Data are expressed as mean ± SDs or median (interquartile range) as appropriate. C group = control group; K group = esketamine group; QoR-15 = The 15-item Quality of Recovery Scale; $T_{before}$: before surgery; $T_{24h}$: 24 h after surgery; $T_{72h}$: 72 h after surgery.

$^*$Compared with before surgery, $P < 0.05$.

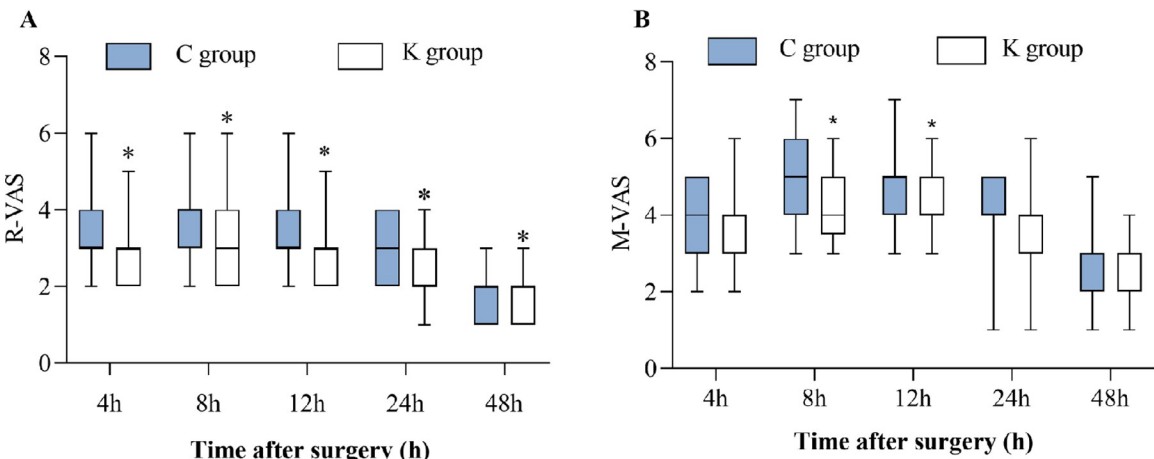

**Fig 2. Box plots of raw data for VAS pain scores at rest (VAS-R) and on coughing (VAS-M) at 4, 8, 12, 24 and 48 h postoperatively.** Data are median (central line), interquartile range (box margins), and adjacent values (whiskers). C group = control group; K group = esketamine group. *Compared with C group, $P < 0.05$.

Stark et al. [21] developed the QoR-15 by screening the most representative psychometrically performing items from each of the five dimensions (emotional state, physical comfort, psychological support, physical independence, and pain) in the QoR-40. The QoR-15 is a multidimensional, patient-centered measure that has been thoroughly evaluated and psychometrically validated [22]. When compared to the QoR-40, the QoR-15 is less time-consuming, easier to use, and more practical in clinical practice; is the most widely used tool to assess patient's quality of recovery [17]. As a result, in our study, the QoR-15 scale was used to assess the effect of esketamine on postoperative quality of recovery.

Cheng X et al. [23] discovered that intravenous esketamine administered during surgery significantly improved postoperative recovery quality in patients undergoing video-assisted thoracic surgery (VATS). The present study found that the most significant differences in QoR-15 between the K and C groups were in the emotional state, physical comfort, and physical independence dimensions, implying that intraoperative intravenous esketamine is beneficial to promote early postoperative recovery in patients undergoing colorectal cancer

**Table 3. Intraoperative data between two groups.**

| | C group (n = 43) | K group (n = 45) | $t/\chi^2$/Z-value | P- value |
|---|---|---|---|---|
| Surgical duration (min) | 175.6 ± 52.0 | 174.4 ± 48.8 | 0.114 | 0.909 |
| Remifentanil dosage (μg) | 2088.8 ± 770.8 | 1845.2 ± 569.1 | 1.681 | 0.097 |
| Cisatracurium dosage (mg) | 28.0 (24.0,32.0) | 27.0 (26.0,29.5) | - 0.591 | 0.555 |
| Application vasoactive drugs, n (%) | 35 (81.4) | 23 (51.1) | 8.975 | 0.003 |
| Infusion amount (ml) | 2300 (2100,2700) | 2600 (2250,2850) | - 1.870 | 0.061 |
| Urine amount (ml) | 300 (150,450) | 400 (250,450) | - 1.855 | 0.064 |
| Blood loss (ml) | 50 (30,50) | 50 (25,50) | - 1.251 | 0.211 |
| Respiratory recovery time (min) | 13.6 ± 4.8 | 11.8 ± 3.5 | 2.059 | 0.042 |
| Extubation time (min) | 18.9 ± 5.7 | 17.6 ± 4.4 | 1.240 | 0.218 |
| Emergence agitation, n(%) | 6 (14.0) | 4 (8.9) | 0.170 | 0.680[a] |

Data are presented as mean ± SDs, the number of patients (%), or median (interquartile range) as appropriate. C group = control group; K group = esketamine group.
[a]is the corrected chi-square test.

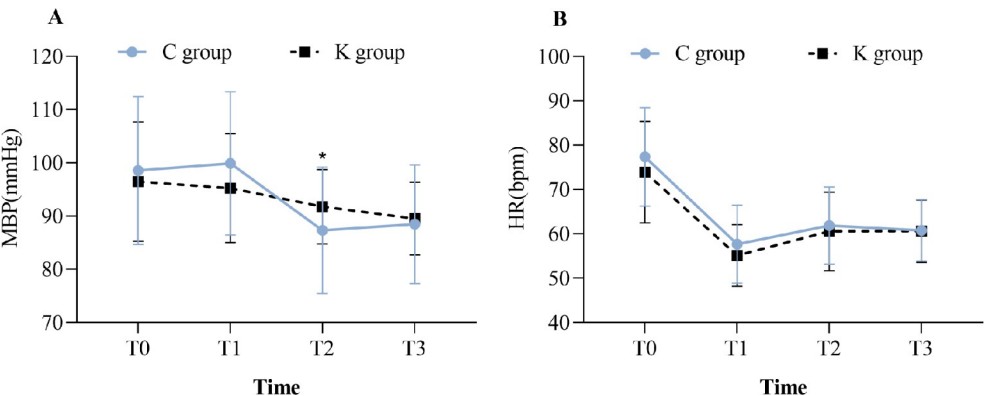

**Fig 3. Intraoperative hemodynamic outcomes.** Data are expressed as mean ± SDs. C group = control group; K group = esketamine group; $T_0$: before induction; $T_1$: after carbon dioxide pneumoperitoneum establishment; $T_2$: after carbon dioxide pneumoperitoneum closure; $T_3$: at the end of surgery. *Compared with C group, $P < 0.05$.

laparoscopic radical resection versus normal saline. There are several reasons about esketamine for improving patient's recovery. First, esketamine may reduce postoperative acute pain score and opioid consumption, as well as opioid-related adverse reactions, thereby hastening patient's recovery [24]. Second, esketamine raises the levels of neurotransmitters such as dopamine in the ventral striatum and caudate nucleus, which in turn stimulates limbic structures, resulting in a good mood and comfort in patients [25]. A previous study found that the minimal clinically important difference (MCID) in the QoR-15 scale was 8.0 [18]. In this study, intraoperative intravenous esketamine resulted in an increase of 8.72 points of the total QoR-15 scores at 24 h after surgery. As the difference between the two groups exceeded the MCID, intraoperative intravenous esketamine significantly improved early postoperative recovery in patients undergoing laparoscopic radical resection of CRC and had some clinical significance. In contrast, Zhao Z et al. [26] found that intraoperative low-dose ketamine did not improve overall quality of recovery in breast cancer surgery patients on postoperative day 1. The disparity in studies could be attributed to differences in patient demographics, drug types, and surgical traumas.. Furthermore, the use of other analgesics, such as nonsteroidal anti-inflammatory drugs and glucocorticoids, which may be able to block NMDA receptors and thus increases

**Table 4. Postoperative data between two groups.**

| | C group (n = 43) | K group (n = 45) | $t/\chi^2/Z$-value | P- value |
|---|---|---|---|---|
| Rescue analgesia in PACU, n (%) | 17 (39.5) | 9 (20.0) | 4.031 | 0.045 |
| Times of effective PCIA press at 24 h | 4.0 (3.0,6.0) | 1.0 (0.0,2.5) | - 5.510 | 0.000 |
| Times of effective PCIA press between 24 h and 48 h | 1.0 (1.0,2.0) | 0.0 (0.0,0.0) | - 5.574 | 0.000 |
| Oxycodone consumption at 24 h (mg) | 17.3 ± 2.3 | 15.7 ± 2.6 | 3.115 | 0.002 |
| Oxycodone consumption between 24 h and 48 h (mg) | 15.0 ± 1.9 | 14.4 ± 1.9 | 1.363 | 0.176 |
| Additional analgesic drugs in the ward, n (%) | 17 (39.5) | 9 (20) | 4.031 | 0.045 |
| Intestinal function recovery time (h) | 36.3 ± 8.8 | 27.8 ± 2.8 | 6.013 | 0.000 |
| Time of first eating or drinking (h) | 20.4 ± 2.2 | 19.6 ± 3.6 | 1.150 | 0.254 |
| Ambulation time (h) | 23.9 ± 3.1 | 21.2 ± 3.5 | 3.842 | 0.000 |
| Hospital stay durations (d) | 10 (9,13) | 10 (8.5,11.0) | - 1.083 | 0.279 |

Data are presented as mean ± SDs, number of patients (%), or median (interquartile range) as appropriate. C group = control group; K group = esketamine group; PACU = post anesthesia care unit; PCIA = patient-controlled intravenous analgesia.

**Table 5. Postoperative adverse events between two groups.**

|  | C group (n = 43) | K group (n = 45) | $\chi^2$-value | P- value |
|---|---|---|---|---|
| PONV | 15 (34.9) | 10 (22.2) | 1.733 | 0.188 |
| Use of antiemetic drugs | 2 (4.7) | 2 (4.4) | 0.000 | 1.000[a] |
| Dizziness | 6 (14.0) | 8 (17.8) | 0.240 | 0.624 |
| Hallucination | 0 (0.0) | 1 (2.2) |  | 1.000[b] |
| Rash | 0 (0.0) | 1 (2.2) |  | 1.000[b] |

Data are presented as patients (%) as appropriate. C group = control group; K group = esketamine group; PONV = postoperative nausea and vomiting.

[a]is the corrected chi-square test;

[b]is the Fisher's exact test.

differences in recovery quality between groups, was restricted in our study. More research is required to validate these hypotheses.

Postoperative pain and delayed recovery of intestinal function are important factors influencing patients' ability to recover quickly after surgery [27]. This present study showed that the VAS-R scores and VAS-M scores at different time points after surgery, the incidence of rescue analgesia in PACU and additional analgesics in the ward, times of effective PCIA press and oxycodone consumption at 24 h after surgery were significantly lower in the K group, indicating that intraoperative intravenous esketamine could provide good postoperative analgesia for patients undergoing laparoscopic radical resection of CRC. This was consistent with previous research findings [24, 28–30]. However, the pain dimension scores of QoR-15 were not significantly different between groups, which could be because the QoR-15 scale focuses on pain frequency while the VAS is more concerned with pain severity. Our study also found that the intestinal function recovery time was shorter in K group, indicating that intraoperative intravenous esketamine, when compared to normal saline, can promote postoperative intestinal function recovery in patients undergoing laparoscopic radical resection of CRC. This was most likely due to esketamine's ability to reduce opioid-related side effects [31].

Opioids are known cause respiratory depression [32]. There was no statistical difference in the use of muscle relaxants and remifentanil between the two groups in our study, but the respiratory recovery time was shorter in the K group; which was likely due to esketamine's ability to reduce opioid-induced respiratory depression by increasing ventilatory $CO_2$ sensitivity [33]. Moreover, esketamine increases sympathetic excitability while decreasing neuronal norepinephrine reuptake [34]. Increased norepinephrine levels in synapses stimulate respiratory activity and increase ventilatory $CO_2$ responsiveness [35, 36]. Furthermore, hydroxynorketamine, an esketamine metabolite, is an agonist of the AMPA (α-amino-3-hydroxy-5-methyl-4-isoxazole-propionic acid) receptor [37]. The AMPA receptors are expressed in brain-stem respiratory networks, such as the pre-Bötzinger complex, a brain area involved in respiratory rhythm generation, and play an important role in maintaining respiratory drive [38].

A previous study found that low-dose esketamine can increase hemodynamic stability during induction of anesthesia in elderly patients undergoing total knee arthroplasty [39]. In our study, the use of vasoactive drugs was lower in the K group, which was likely to esketamine's indirect effect on the cardiovascular system by stimulating the sympathetic nervous system, reducing hemodynamic fluctuation during anesthesia in patients with CRC. According to some studies, esketamine does not reduce the occurrence of PONV [23, 26]. A review, on the other hand, concluded that perioperative intravenous ketamine can reduce PONV [40]. The occurrence of PONV was not statistically different between the groups in our study, which could be attributed to the routine administration of antiemetics prior to the end of the surgery,

as well as postoperative patient-controlled intravenous analgesia. Intraoperative intravenous esketamine improved postoperative recovery quality and shortened ambulation time, but postoperative hospital stay durations were comparable between groups. One possible explanation for this outcome was that the criteria for patient discharge from the hospital was inconsistent. As a result, more clinical trials are required for further research exploration.

There were some limitations in this study. Firstly, the patients in our study were ASA I or II, relatively healthy, and had an upper age limit. Further research should be conducted on elderly patients with ASA III or IV to confirm the findings' generalizability. Secondly, we only looked at the effect of esketamine on early recovery quality, and patients were not followed up for a long time after surgery. A previous study found a link between the quality of recovery in the hospital and the quality of life three months after surgery [41]. Therefore, the current study's findings have some implications for the patients' long-term health. Furthermore, the sample size calculated based on the total QoR-15 score may have been insufficient for secondary outcomes observation, and only one dose of esketamine was tested. A previous study showed that esketamine as an adjunct can reduce opioid consumption after major lumbar fusion in a dose-dependent manner [42]. Determining the optimal esketamine dose and route of administration requires further research.

In conclusion, intraoperative intravenous low-dose esketamine can improve the early postoperative quality of recovery, reduce pain scores and opioid consumption in patients undergoing laparoscopic radical resection of CRC without increasing adverse reactions. Future clinical trials with other types of surgery will be required to investigate the long-term effects of esketamine on patients' recovery.

## Supporting information

**S1 File. CONSORT 2010 checklist.**
(DOC)

**S2 File. Study protocol (Original Chinese).**
(DOCX)

**S3 File. Study protocol (English).**
(DOCX)

## Acknowledgments

The authors express gratitude to our anesthesiology and colorectal surgery teachers for their cooperation in facilitating this research. We also thank the anesthesia nurses for their kind assistance.

## Author Contributions

**Conceptualization:** Ying Xu, Long He, Yanqiu Ai.

**Data curation:** Ying Xu, Shaoxuan Liu.

**Formal analysis:** Ying Xu, Shaoxuan Liu.

**Funding acquisition:** Long He, Yanqiu Ai.

**Investigation:** Ying Xu, Long He, Chaofan Zhang.

**Methodology:** Ying Xu, Long He, Yanqiu Ai.

**Project administration:** Ying Xu.

**Supervision:** Ying Xu, Long He, Yanqiu Ai.

**Visualization:** Ying Xu, Chaofan Zhang.

**Writing – original draft:** Ying Xu, Long He.

**Writing – review & editing:** Long He, Yanqiu Ai.

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
