## [Decision Letter · Decision Letter 0]

23 Jan 2023

PONE-D-22-33531Intraoperative intravenous low-dose esketamine improves quality of early recovery after laparoscopic radical resection of colorectal cancer: A prospective, randomized controlled trialPLOS ONE

Dear Dr. Ai,

Thank you for submitting your manuscript to PLOS ONE. After careful consideration, we feel that it has merit but does not fully meet PLOS ONE’s publication criteria as it currently stands. Therefore, we invite you to submit a revised version of the manuscript that addresses the points raised during the review process.

We look forward to receiving your revised manuscript.

Kind regards,

Ahmed Mohamed Ibrahim Hasanin

Academic Editor

PLOS ONE

Journal Requirements:

"This work was supported by the National Natural Science Foundation of China (grant number: 82001189) to L.H. and the Henan Science and Technology project (grant number: 201911225) to Y.A."

Reviewers' comments:

Reviewer's Responses to Questions

**Comments to the Author**

1. Is the manuscript technically sound, and do the data support the conclusions?

Reviewer #1: Yes

Reviewer #2: Yes

Reviewer #3: Yes

2. Has the statistical analysis been performed appropriately and rigorously? 

Reviewer #1: Yes

Reviewer #2: Yes

Reviewer #3: No

3. Have the authors made all data underlying the findings in their manuscript fully available?

Reviewer #1: No

Reviewer #2: Yes

Reviewer #3: Yes

4. Is the manuscript presented in an intelligible fashion and written in standard English?

Reviewer #1: Yes

Reviewer #2: No

Reviewer #3: Yes

5. Review Comments to the Author

Reviewer #1: Comments on PONE-D-22-33531

Thank you for inviting me to revise this manuscript. The authors discussed a clinically relevant topic. The manuscript is generally well written, but I have some comments.

Abstract

- VAS was mentioned without identification

- You mentioned group k and group C in the results without identification to which group they referred to. Please identify the group name in the method part of the abstract e.g.: “were randomized to Esketamine (K group) and non-eskatamine groups (C group)” however; using a self-explaining name for the group is better

- No need to mention all the study outcomes in the abstract specially when their results were not mentioned in the results section of the abstract

- The conclusion in the abstract should reflect the information mentioned in it, and since you did not mention any results related to adverse events in the results part, you should not mention that the drug “and did not increase postoperative adverse reaction”

- Please change colorectal patients to patients with colorectal cancer

Introduction

- In the first paragraph, you need to explain the operative related complication that could be improved by the study drug. Otherwise, it is better to remove the sentence starting with “However, the establishment…” as the role of the study drug is not to improve or impact those complications. You may consider other complications, such as inadequate pain management and use of opioids, which could be affected by the study drug.

- In the second paragraph, mention some of the adverse effect of ketamine in comparison to the esketamine.

- In the same paragraph, you should break down the last sentence.

Methods

- The blinding is not clear, you described the rate in the esketamine group as 0.25mg/kg of esketamine as bolus, and then a rate of 0.12mg/kg/h, and that equal volume of normal saline was infused in the C group. please clarify how is equal volume was calculated

- In the secondary outcomes please remove “etc” and enumerate the recorded complications.

Sample size

Please provide more details regarding the assumed SD for the sample size calculation and the used program, ref 18 is not relevant to your study population

Statistical analysis

Please change:

“A Student t-test” to “the Student t-test”

“Datas” to Data

(interquartile range) to (quartiles)

Add how the normality of data was checked

Results

In the first paragraph last sentence please edit to “were not significantly different”

Discussion

P 9 third paragraph: S-ketamine to esketamine

Some grammatical mistakes: “leaded” should be “led”

Reviewer #2: Title: Intraoperative intravenous low-dose esketamine improves quality of early recovery after laparoscopic radical resection of colorectal cancer: A prospective, randomized controlled trial .

Thank you for the chance of reviewing this interesting manuscript. This retrospective study aimed to assess the effects of intraoperative intravenous low-dose esketamine on the quality of early recovery in patients undergoing laparoscopic radical resection of CRC using the QoR-15 scale. The study design has exciting clinical guiding value. Especially for the management of multimodal postoperative analgesia, it has a very strong research significance. However, in order to achieve the purpose of the study, there are significant methodological problems. I have the following comments:

1.The article's writing quality has to be further enhanced because it does not adhere to the magazine's standards for statistical analysis methods, tables, charts, and other elements.

2.To make the text more readable, the authors should modify the language. I suggest the authors hire a qualified native English editor.

3.To ensure that readers fully grasp what the researchers looked at, the technical information should be enlarged and explained.

4.Page 1 Methods, VAS, PACU, etc., Abbreviations should be noted in full English whenever they first appear.

5.Please update the ‘Methods’ section of the abstract to include details of the methods, results.

6.The grouping of methods and results in the abstract is not clearly described.

7.Please update the ‘Introduction’ section of the therefore.

8.Why was the age range for the study's participants set at 18 to 65 years old? I don't think this age structure makes sense; if you can, please cite some references.

9.‘Measures related to the blood specimens in the study protocol were not shown in the manuscript because of improper storage of the blood specimens. ’ ? It appeared out of nowhere. I don't comprehend.

10.Please provide information to the "Sample size calculation" section.

11.Figure 1, Fig 1......, Verify the syntax, unify the names, and review the entire document.

12.Table 1, etc., We preserve how many decimal places? Please consult the literature; all statistical findings must adhere to a single standard.

13.Table 2, 3, 4, It is appropriate for the appropriate position.

Thanks

Reviewer #3: An interesting clinical trial that left me with a lot of questions:

1. You cite reference 18 as giving the sample size, but it's not clear to me that this trial had the same primary outcome, the same patient population, and the same variability that you had in your study. Can you justify using someone else's sample size computation? And if they already did the same study you did, why did you have to redo it?

2. So many p-values, and no adjustment for multiple testing. Yet you say 0.048 is significant. Without an adjustment for multiple testing, I don't believe your "significance" statements.

3. How do I know your sample size was adequate? It would be helpful to add to the discussion whether the assumptions underlying these computations were actually realized in the trial, but then you don't state the assumptions because they are taken from a different paper.

4. Anytime you use repeated measures ANOVA, you need to give the assumptions of the model, and how you verified those assumptions (e.g., diagnostics).

6. PLOS authors have the option to publish the peer review history of their article (what does this mean?). If published, this will include your full peer review and any attached files.

Reviewer #1: **Yes: **Maha Mostafa

Reviewer #2: **Yes: **Yongtao Sun

Reviewer #3: No

---

## [Author Response · Author response to Decision Letter 0]

13 Feb 2023

Dear Editor:

 Thank you for your useful comments on our manuscript. We wish to give a sincere gratitude to referees for reviewing our paper carefully. We apologize for any inconveniences caused by these errors. We have modified the manuscript accordingly, and the specific modifications are listed point by point below.

 Reviewer: 1

Abstract

- VAS was mentioned without identification 

Thank you for the suggestion. This section has been removed due to the third comment “No need to mention all the study outcomes”. However, VAS was mentioned with identification on the page 3 “The visual analogue scale (VAS) was used to assess pain intensity;”. 

- You mentioned group k and group C in the results without identification to which group they referred to. Please identify the group name in the method part of the abstract e.g.: “were randomized to Esketamine (K group) and non-eskatamine groups (C group)” however; using a self-explaining name for the group is better

Thank you for the suggestion. We have corrected it in the method part of the abstract : “were randomly assigned to either the esketamine (K group) or non-eskatamine (C group) group.”.

- No need to mention all the study outcomes in the abstract specially when their results were not mentioned in the results section of the abstract

Thank you for the suggestion. Secondary outcomes in our study has been removed in the abstract.

- The conclusion in the abstract should reflect the information mentioned in it, and since you did not mention any results related to adverse events in the results part, you should not mention that the drug “and did not increase postoperative adverse reaction”

Thank you for the suggestion. We added that “The postoperative adverse events were comparable between the two groups.” in the result part.

- Please change colorectal patients to patients with colorectal cancer

Thank you very much for the suggestion. We have corrected it to patients with colorectal cancer.

Introduction

- In the first paragraph, you need to explain the operative related complication that could be improved by the study drug. Otherwise, it is better to remove the sentence starting with “However, the establishment…” as the role of the study drug is not to improve or impact those complications. You may consider other complications, such as inadequate pain management and use of opioids, which could be affected by the study drug.

Thank you for the suggestion. We have revised the sentence to “However, pain after laparoscopic surgery is a common issue that may interfere with discharge and recovery.”.

- In the second paragraph, mention some of the adverse effect of ketamine in comparison to the esketamine. 

Thank you for the suggestion. We have added this sentence “Because of the dose-dependent side effects of ketamine, esketamine has a lower likelihood of adverse reactions (such as psychedelic effects).”.

- In the same paragraph, you should break down the last sentence.

Thank you for the suggestion. We have broken down the last sentence that “The 15-item Quality of Recovery scale (QoR-15) is a patient-centered outcome measure used to assess patients’ health status following surgery and anesthesia. Furthermore, the QoR-15 is dependable, valid and has high clinical acceptability.”.

Methods

- The blinding is not clear, you described the rate in the esketamine group as 0.25mg/kg of esketamine as bolus, and then a rate of 0.12mg/kg/h, and that equal volume of normal saline was infused in the C group. please clarify how is equal volume was calculated

Thank you for the suggestion. We have clarified this with “Esketamine, 50 mg, was diluted to a total of 50 ml with normal saline solution. Both esketamine and normal saline were drawn into identical 50-ml syringes by the same anesthesia nurse.”.

- In the secondary outcomes please remove “etc” and enumerate the recorded complications.

Thank you for the suggestion. We have removed “etc” and enumerate the recorded complications “pruritis”.

Sample size

Please provide more details regarding the assumed SD for the sample size calculation and the used program, ref 18 is not relevant to your study population 

Thank you for the suggestion. We have provided more details on the sample size calculation :“The primary outcome of this study was the quality of recovery score (QoR-15) at 24 h postoperatively. PASS 15.0 was used to calculate the estimated sample size based on the primary outcome. The minimal clinically important difference (MCID) in the QoR-15 scale was 8.0. According to previous research and preliminary findings, the total QoR-15 score at 24 h postoperatively in the esketamine group was 114 ± 16, compared to 102 ± 22 in the control group. A sample size of 42 subjects in each group was calculated using the preliminary study data, assuming Type Ⅰ error (α)＝0.05 and Type Ⅱ error (β)＝0.2. To allow for a dropout rate of 10%, we finally enrolled 46 subjects in each group in this study.”.

Statistical analysis

Please change:

“A Student t-test” to “the Student t-test”

Thank you for the suggestion. We have corrected it to the Student’s t-test.

“Datas” to Data

Thank you for the suggestion. We have corrected it.

(interquartile range) to (quartiles)

Thank you for the suggestion. We have corrected it to (quartiles).

Add how the normality of data was checked

Thank you for the suggestion. It’s in the second sentence of this part “The Shapiro-Wilk test was used to test the data’s normality.”

Results

In the first paragraph last sentence please edit to “were not significantly different”

Thank you for the suggestion. We have corrected it to “were not significantly different”

Discussion

P 9 third paragraph: S-ketamine to esketamine

Thank you very much for the suggestion. We have corrected it to esketamine.

Some grammatical mistakes: “leaded” should be “led”

Thank you very much for the suggestion. We have corrected it.

Reviewer: 2

1.The article's writing quality has to be further enhanced because it does not adhere to the magazine's standards for statistical analysis methods, tables, charts, and other elements.

 Thank you for the suggestion. We have made appropriate changes according to the magazine’s standards.

2.To make the text more readable, the authors should modify the language. I suggest the authors hire a qualified native English editor.

 Thank you for the suggestion. We have hired one or more of the highly qualified native English speaking editors at Springer Nature Author Services (SNAS) to modify the language. The Editing Certificate has been uploaded as an attachment.

3.To ensure that readers fully grasp what the researchers looked at, the technical information should be enlarged and explained.

 Thank you for the suggestion. We have made some changes to this.

4.Page 1 Methods, VAS, PACU, etc., Abbreviations should be noted in full English whenever they first appear.

 Thank you for the suggestion. This section has been removed because we made appropriate changes to the abstract. And, abbreviations have been noted in full English in the Methods part on Page 4. 

5.Please update the ‘Methods’ section of the abstract to include details of the methods, results.

 Thank you for the suggestion. We have updated the ‘Methods’ section of the abstract “This was a prospective, randomized controlled trial. Ninety-two patients undergoing laparoscopic radical resection of colorectal cancer were randomly assigned to either the esketamine (K group) or non-eskatamine (C group) group. After anesthesia induction, a loading dose of 0.25 mg/kg was administered, followed by continuous infusion at a rate of 0.12 mg/kg/h until the end of surgery in the K group. In the C group, an equivalent volume of normal saline was infused. The primary outcome was quality of recovery at 24 h after surgery, as measured by the Quality of Recovery-15 (QoR-15) scale. The QoR-15 was evaluated at three timepoints: before (Tbefore), 24 h (T24h) and 72 h (T72h) after surgery. ”.

6.The grouping of methods and results in the abstract is not clearly described.

 Thank you very much for the suggestion. We have corrected it. The grouping of methods in the abstract is described as “Ninety-two patients undergoing laparoscopic radical resection of colorectal cancer were randomly assigned to either the esketamine (K group) or non-eskatamine (C group) group.”. And, the results in the abstract is described as “A total of 88 patients completed this study. The total QoR-15 scores in K group (n=45) were higher than in the C group (n=43) at 24 h: 112.33 ± 8.79 vs. 103.93 ± 9.03 (P = 0.000) and at 72h: 118.73 ± 7.82 vs. 114.79 ± 7.98 (P = 0.022). However, the differences between the two groups only had clinical significance at 24h after surgery. Among the five dimensions of the QoR-15, physical comfort (P = 0.003) ,emotional state (P = 0.000), and physical independence (P = 0.000) were significantly higher at 24h in the K group, and physical comfort (P = 0.048) was higher at 72h in the K group. The postoperative adverse events were comparable between the two groups.”.

7.Please update the ‘Introduction’ section of the therefore.

 Thank you for the suggestion. We have made some appropriate changes to the ‘Introduction’ section.

8.Why was the age range for the study's participants set at 18 to 65 years old? I don't think this age structure makes sense; if you can, please cite some references.

 Thank you for the suggestion. We are very sorry for your confusion. According to the medicine specification of esketamine (SFDA Approval No.: H20193336, Jiangsu Hengrui Pharmaceutical Co Ltd), the use of esketamine is not clear in children and elderly patients. Therefore, the age range for the study's participants was set at 18 to 65 years old.

9.‘Measures related to the blood specimens in the study protocol were not shown in the manuscript because of improper storage of the blood specimens. ’ ? It appeared out of nowhere. I don't comprehend.

 Thank you for the suggestion. We are very sorry for your confusion. The sentence is to illustrate the item 6b ‘Any changes to trial outcomes after the trial commenced, with reasons’ of the CONSORT 2010 checklist. Due to improper storage of blood samples, the results associated with them could not be shown in the manuscript. We have removed it.

10.Please provide information to the "Sample size calculation" section.

 Thank you for the suggestion. We have provided more information on the sample size calculation :“The primary outcome of this study was the quality of recovery score (QoR-15) at 24 h postoperatively. PASS 15.0 was used to calculate the estimated sample size based on the primary outcome. The minimal clinically important difference (MCID) in the QoR-15 scale was 8.0. According to previous research and preliminary findings, the total QoR-15 score at 24 h postoperatively in the esketamine group was 114 ± 16, compared to 102 ± 22 in the control group. A sample size of 42 subjects in each group was calculated using the preliminary study data, assuming Type Ⅰ error (α)＝0.05 and Type Ⅱ error (β)＝0.2. To allow for a dropout rate of 10%, we finally enrolled 46 subjects in each group in this study.”. 

11.Figure 1, Fig 1......, Verify the syntax, unify the names, and review the entire document.

 Thank you very much for the suggestion. We have verified the the syntax, unified the names, and reviewed the entire document.

12.Table 1, etc., We preserve how many decimal places? Please consult the literature; all statistical findings must adhere to a single standard.

 Thank you very much for the suggestion. We have consulted the literature on decimal places and modified it.

13.Table 2, 3, 4, It is appropriate for the appropriate position.

 Thank you very much for the suggestion.

Reviewer: 3

1. You cite reference 18 as giving the sample size, but it's not clear to me that this trial had the same primary outcome, the same patient population, and the same variability that you had in your study. Can you justify using someone else's sample size computation? And if they already did the same study you did, why did you have to redo it?

 Thank you for the suggestion. We apologize for the inappropriate expression we made in the manuscript. Our study had the same primary outcome as reference 18, but the patient population and primary intervention are different. We have provided more details on the sample size calculation :“The primary outcome of this study was the quality of recovery score (QoR-15) at 24 h postoperatively. PASS 15.0 was used to calculate the estimated sample size based on the primary outcome. The minimal clinically important difference (MCID) in the QoR-15 scale was 8.0. According to previous research and preliminary findings, the total QoR-15 score at 24 h postoperatively in the esketamine group was 114 ± 16, compared to 102 ± 22 in the control group. A sample size of 42 subjects in each group was calculated using the preliminary study data, assuming Type Ⅰ error (α)＝0.05 and Type Ⅱ error (β)＝0.2. To allow for a dropout rate of 10%, we finally enrolled 46 subjects in each group in this study.”.

2. So many p-values, and no adjustment for multiple testing. Yet you say 0.048 is significant. Without an adjustment for multiple testing, I don't believe your "significance" statements.

Thank you for the suggestion. We apologiz for the inappropriate expression. What we're trying to say is that 0.048 is a statistical difference and physical comfort (42.31 ± 4.55 vs. 40.26 ± 5.04, P = 0.048) was higher at 72 h in the K group. We have corrected it.

3. How do I know your sample size was adequate? It would be helpful to add to the discussion whether the assumptions underlying these computations were actually realized in the trial, but then you don't state the assumptions because they are taken from a different paper.

 Thank you for the suggestion. In response to your question, we have conducted the post hoc analysis. Based on the mean and standard deviation (SD) of the two groups, the effect size dCohen calculated was 0.943 by the following website: https://www.psychometrica.de/effect_size.html Next, G*Power 3.1.9.7 was used to calculate the Power (1－βerr prob), which was 0.994. It is greater than 0.8. Therefore, our sample was adequate. 

4. Anytime you use repeated measures ANOVA, you need to give the assumptions of the model, and how you verified those assumptions (e.g., diagnostics).

 Thank you for the suggestion. We apologize for the inaccurate expression. We have corrected it to “Data with normal distribution at multiple time points were analyzed using repeated measures analysis of variance (ANOVA).”.

Editor comments:

 Thank you for the suggestion. Our manuscript has been revised according to PLOS ONE's style requirements.

"This work was supported by the National Natural Science Foundation of China (grant number: 82001189) to L.H. and the Henan Science and Technology project (grant number: 201911225) to Y.A."

 Thank you for the suggestion. The funders had no role in study design, data collection and analysis, decision to publish, or preparation of the manuscript.

 Thank you for the suggestion. The captions for our Supporting Information files have been included at the end of our manuscript. And we have updated any in-text citations to match accordingly.

---

## [Decision Letter · Decision Letter 1]

24 Mar 2023

PONE-D-22-33531R1Intraoperative intravenous low-dose esketamine improves quality of early recovery after laparoscopic radical resection of colorectal cancer: A prospective, randomized controlled trialPLOS ONE

Dear Dr. Ai,

Thank you for submitting your manuscript to PLOS ONE. After careful consideration, we feel that it has merit but does not fully meet PLOS ONE’s publication criteria as it currently stands. Therefore, we invite you to submit a revised version of the manuscript that addresses the points raised during the review process.

We look forward to receiving your revised manuscript.

Kind regards,

Ahmed Mohamed Ibrahim Hasanin

Academic Editor

PLOS ONE

Additional Editor Comments:

There are still major problem with the statistics, namely the sample size calculation. The authors changed the source of the data of their sample size calculation to a pilot study instead of a previous study. The authors need to clarify why they changed the source of the data. Do they think that the original calculation was not correct and they performed a post-hoc calculation?

Reviewers' comments:

Reviewer's Responses to Questions

**Comments to the Author**

1. If the authors have adequately addressed your comments raised in a previous round of review and you feel that this manuscript is now acceptable for publication, you may indicate that here to bypass the “Comments to the Author” section, enter your conflict of interest statement in the “Confidential to Editor” section, and submit your "Accept" recommendation.

Reviewer #1: (No Response)

Reviewer #2: (No Response)

Reviewer #3: (No Response)

2. Is the manuscript technically sound, and do the data support the conclusions?

Reviewer #1: Yes

Reviewer #2: Partly

Reviewer #3: No

3. Has the statistical analysis been performed appropriately and rigorously? 

Reviewer #1: I Don't Know

Reviewer #2: No

Reviewer #3: No

4. Have the authors made all data underlying the findings in their manuscript fully available?

Reviewer #1: No

Reviewer #2: Yes

Reviewer #3: Yes

5. Is the manuscript presented in an intelligible fashion and written in standard English?

Reviewer #1: Yes

Reviewer #2: Yes

Reviewer #3: Yes

6. Review Comments to the Author

Reviewer #1: Abstract

- Enumerate the adverse events in the method section of the abstract as you still mentioning it in the results and conclusion of the abstract for example “Other outcome include the incidence of postoperative adverse events (nausea, vomiting, dizziness, hallucination and pruritis)”

Introduction

- Please change colorectal patients to patients with colorectal cancer

Methods

- Change muscle relaxation to neuromuscular blockade.

The response to reviewer 3 point #2 is not clear

Reviewer #2: (No Response)

Reviewer #3: Many of my statistical comments were either not understood or not responded to adequately. The authors now admit that the paper that they based their sample size on is not relevant to this study, and now redo the sample size analysis from scratch. It seems odd to me that this new analysis gives the sample size used in the study. Typically, a study cannot be remediated when the design is incorrect. Secondly, the authors now conduct a post-hoc analysis using completely DIFFERENT techniques than were specified in the original analysis. The study was never powered for a correlation, it was powered by a difference in means. My comments on statistical significance and multiple testing was not addressed adequately. Distributional assumptions were not verified with diagnostics.

7. PLOS authors have the option to publish the peer review history of their article (what does this mean?). If published, this will include your full peer review and any attached files.

Reviewer #1: No

Reviewer #2: **Yes: **Yongtao Sun

Reviewer #3: No

---

## [Author Response · Author response to Decision Letter 1]

30 Mar 2023

Dear Editors:

 Thank you for your useful comments on our manuscript. We wish to give a sincere gratitude to referees for reviewing our paper carefully. We apologize for any inconveniences caused by these errors. We have modified the manuscript accordingly, and the specific modifications are listed point by point below. The responses to the reviewers’ comments are marked using yellow highlight and presented following.

 We would love to thank you for allowing us to resubmit a revised copy of the manuscript and we highly appreciate your time and consideration.

Best regards

Sincerely yours

Additional Editor Comments:

There are still major problem with the statistics, namely the sample size calculation. The authors changed the source of the data of their sample size calculation to a pilot study instead of a previous study. The authors need to clarify why they changed the source of the data. Do they think that the original calculation was not correct and they performed a post-hoc calculation?

Response: We are very grateful to your comments for the manuscript. Your suggestion are very important and have important guiding significance for my thesis writing and scientific research work. We are very sorry for your misunderstanding caused by the inappropriate expression we made in the manuscript. We didn’t change the source of the data of our sample size calculation. We calculated the sample size by combining the previous study and the preliminary findings. First, we determined the primary outcome based on the previous study, and then we determined the sample size based on the results of the pilot study. 

Thanks again for your advice and I hope to learn more from you.

Reviewer: 1

Abstract

- Enumerate the adverse events in the method section of the abstract as you still mentioning it in the results and conclusion of the abstract for example “Other outcome include the incidence of postoperative adverse events (nausea, vomiting, dizziness, hallucination and pruritis)”

Response: Thank you for the suggestion. Your suggestion are very important and have important guiding significance for my thesis writing and scientific research work. The adverse events in the abstract has been removed.

Thanks again for your advice and I hope to learn more from you.

Introduction

- Please change colorectal patients to patients with colorectal cancer

Response: Thank you very much for the suggestion. Your suggestion are very important and have important guiding significance for my thesis writing and scientific research work. We have corrected it to patients with colorectal cancer. Thanks again for your advice and I hope to learn more from you.

Methods

- Change muscle relaxation to neuromuscular blockade.

Response: Thank you very much for the suggestion. Your suggestion are very important and have important guiding significance for my thesis writing and scientific research work. We have corrected it to neuromuscular blockade.

Thanks again for your advice and I hope to learn more from you.

 The response to reviewer 3 point #2 is not clear

reviewer 3 point #2: So many p-values, and no adjustment for multiple testing. Yet you say 0.048 is significant. Without an adjustment for multiple testing, I don't believe your "significance" statements.

Response: Thank you very much for the suggestion. Your suggestion are very important and have important guiding significance for my thesis writing and scientific research work. Data at multiple time points, such as total QoR-15 scores, QoR-15 domains and VAS scores, were analyzed with an adjustment for multiple testing using Bonferroni or LSD, which was not specifically expressed in the statistical analysis part of the manuscript. Please see the figure below for specific statistical results on physical comfort.

Thanks again for your advice and I hope to learn more from you.

 Reviewer: 1

Abstract

- Enumerate the adverse events in the method section of the abstract as you still mentioning it in the results and conclusion of the abstract for example “Other outcome include the incidence of postoperative adverse events (nausea, vomiting, dizziness, hallucination and pruritis)”

Response: Thank you for the suggestion. Your suggestion are very important and have important guiding significance for my thesis writing and scientific research work. The adverse events in the abstract has been removed.

Thanks again for your advice and I hope to learn more from you.

Introduction

- Please change colorectal patients to patients with colorectal cancer

Response: Thank you very much for the suggestion. Your suggestion are very important and have important guiding significance for my thesis writing and scientific research work. We have corrected it to patients with colorectal cancer. Thanks again for your advice and I hope to learn more from you.

Methods

- Change muscle relaxation to neuromuscular blockade.

Response: Thank you very much for the suggestion. Your suggestion are very important and have important guiding significance for my thesis writing and scientific research work. We have corrected it to neuromuscular blockade.

Thanks again for your advice and I hope to learn more from you.

 The response to reviewer 3 point #2 is not clear

reviewer 3 point #2: So many p-values, and no adjustment for multiple testing. Yet you say 0.048 is significant. Without an adjustment for multiple testing, I don't believe your "significance" statements.

Response: Thank you very much for the suggestion. Your suggestion are very important and have important guiding significance for my thesis writing and scientific research work. Data at multiple time points, such as total QoR-15 scores, QoR-15 domains and VAS scores, were analyzed with an adjustment for multiple testing using Bonferroni or LSD, which was not specifically expressed in the statistical analysis part of the manuscript. This file of Response to Reviewers shows the specific statistical result.

Thanks again for your advice and I hope to learn more from you.

Reviewer: 3

1. In the results, statistical methods should be presented in a way that meets the requirements of the journal. For example, whether to keep one decimal point or two decimal points, please unify the standard.

Response: Thank you very much for the suggestion. Your suggestion are very important and have important guiding significance for my thesis writing and scientific research work. We have corrected it to unify the standard for keeping one decimal point.

Thanks again for your advice and I hope to learn more from you.

2. Repeat the question asked the first time. Tables should be presented in three lines.

Response: Thank you very much for the suggestion. Your suggestion are very important and have important guiding significance for my thesis writing and scientific research work. We have corrected the tables to be presented in three lines. And we are very sorry that the first answer was not satisfactory to you as we did not fully understand your question. Here are re-answer to the questions asked the first time.

Thanks again for your advice and I hope to learn more from you.

Point #1: You cite reference 18 as giving the sample size, but it's not clear to me that this trial had the same primary outcome, the same patient population, and the same variability that you had in your study. Can you justify using someone else's sample size computation? And if they already did the same study you did, why did you have to redo it?

Response: Thank you for the suggestion. We apologize for the inappropriate expression we made in the manuscript. Our study had the same primary outcome (the QoR-15 scores at 24 h after surgery) as reference 18, but the patient population and primary intervention are both different. We calculated the sample size by combining the previous study and the preliminary findings. First, we determined the primary outcome based on the reference 18, and then we determined the sample size based on the results of the pilot study.

Thanks again for your advice and I hope to learn more from you.

Point #2: So many p-values, and no adjustment for multiple testing. Yet you say 0.048 is significant. Without an adjustment for multiple testing, I don't believe your "significance" statements.

Response: Thank you very much for the suggestion. Data at multiple time points, such as total QoR-15 scores, QoR-15 domains and VAS scores, were analyzed with an adjustment for multiple testing using Bonferroni or LSD, which was not specifically expressed in the statistical analysis part of the manuscript. This file of Response to Reviewers shows the specific statistical result.

Thanks again for your advice and I hope to learn more from you. 

Point #3: How do I know your sample size was adequate? It would be helpful to add to the discussion whether the assumptions underlying these computations were actually realized in the trial, but then you don't state the assumptions because they are taken from a different paper.

Response: Thank you for the suggestion. We used the PASS 15.0 to calculate the Power (1－β). Based on the mean, standard deviation (SD) and sample size of the two groups, the Power (1－β) calculated in PASS 15.0 was 0.99418. It is greater than 0.8. Therefore, our sample was adequate. This file of Response to Reviewers shows the specific calculation result of the Power (1－β). 

Thanks again for your advice and I hope to learn more from you.

Point #4: Anytime you use repeated measures ANOVA, you need to give the assumptions of the model, and how you verified those assumptions (e.g., diagnostics).

Response: Thank you for the suggestion. In the statistical analysis section of this manuscript, we have shown that “The Shapiro-Wilk test was used to test the data’s normality. Data with normal distribution at multiple time points were analyzed using repeated measures analysis of variance (ANOVA). Data with non-normal distribution at multiple time points were analyzed using the Generalized Estimating Equations (GEE)”. And, the statistical analysis methods of many studies are also expressed in this way, such as the following literature: “Niu Zheng,Gao Xiuxiu,Shi Zeshu et al. Effect of total intravenous anesthesia or inhalation anesthesia on postoperative quality of recovery in patients undergoing total laparoscopic hysterectomy: A randomized controlled trial.[J] .J Clin Anesth, 2021, 73: 110374.” Specific statistical analysis is shown in the figure below.

Thanks again for your advice and I hope to learn more from you.

Reviewer #3: Many of my statistical comments were either not understood or not responded to adequately. The authors now admit that the paper that they based their sample size on is not relevant to this study, and now redo the sample size analysis from scratch. It seems odd to me that this new analysis gives the sample size used in the study. Typically, a study cannot be remediated when the design is incorrect. Secondly, the authors now conduct a post-hoc analysis using completely DIFFERENT techniques than were specified in the original analysis. The study was never powered for a correlation, it was powered by a difference in means. My comments on statistical significance and multiple testing was not addressed adequately. Distributional assumptions were not verified with diagnostics.

Response: We are very grateful to your comments for the manuscript. According with your advice, we amended the relevant part in manuscript. Your suggestion are very important and have important guiding significance for my thesis writing and scientific research work. We are very sorry that we did not adequately understand your statistical comments. We re-answer the questions asked the first time. We don’t redo the sample size analysis from scratch. We are very sorry for your misunderstanding caused by the inappropriate expression we made in the manuscript. We calculated the sample size by combining the previous study and the preliminary findings. First, we determined the primary outcome based on the previous study, and then we determined the sample size based on the results of the pilot study. Secondly, we used the PASS 15.0 to calculate the Power (1－β) to verify the adequacy of the sample size. Bonferroni was used as an adjustment for multiple testing.

We are very sorry that we did not fully understand the sentence “Distributional assumptions were not verified with diagnostics.” We tried to answer it in the Point #4 according to our understanding. Thanks again for your suggestion.

Thank you for your advice. It's very important. Due to your advice, I have found my shortcomings in my current work. I will improve my scientific research level and make more achievements in the future work according to your advice.

Thanks again for your advice and I hope to learn more from you.

---

## [Decision Letter · Decision Letter 2]

19 Apr 2023

PONE-D-22-33531R2

Intraoperative intravenous low-dose esketamine improves quality of early recovery after laparoscopic radical resection of colorectal cancer: A prospective, randomized controlled trial

PLOS ONE

Dear Dr. Ai,

Thank you for submitting your manuscript to PLOS ONE. After careful consideration, we feel that it has merit but does not fully meet PLOS ONE’s publication criteria as it currently stands. Therefore, we invite you to submit a revised version of the manuscript that addresses the points raised during the review process.

We look forward to receiving your revised manuscript.

Kind regards,

Ahmed Mohamed Ibrahim Hasanin

Academic Editor

PLOS ONE

Journal Requirements:

Additional Editor Comments:

Thank you for your nice revision. There is a very minor statistical comment which needs correction before final acceptance of the manuscript. Well done.

Reviewers' comments:

Reviewer's Responses to Questions

**Comments to the Author**

1. If the authors have adequately addressed your comments raised in a previous round of review and you feel that this manuscript is now acceptable for publication, you may indicate that here to bypass the “Comments to the Author” section, enter your conflict of interest statement in the “Confidential to Editor” section, and submit your "Accept" recommendation.

Reviewer #1: (No Response)

Reviewer #2: All comments have been addressed

Reviewer #3: All comments have been addressed

2. Is the manuscript technically sound, and do the data support the conclusions?

Reviewer #1: Yes

Reviewer #2: Yes

Reviewer #3: Partly

3. Has the statistical analysis been performed appropriately and rigorously? 

Reviewer #1: Yes

Reviewer #2: Yes

Reviewer #3: I Don't Know

4. Have the authors made all data underlying the findings in their manuscript fully available?

Reviewer #1: Yes

Reviewer #2: Yes

Reviewer #3: Yes

5. Is the manuscript presented in an intelligible fashion and written in standard English?

Reviewer #1: Yes

Reviewer #2: Yes

Reviewer #3: Yes

6. Review Comments to the Author

Reviewer #1: please state that "Adjustment for multiple testing was done using Bonferroni or LSD" in the statistical analysis section

Reviewer #2: Thanks to the authors of the manuscript for their active and effective revisions. The quality of the manuscript has been greatly improved, basically meeting the requirements for publication. After my own careful examination, i feel that the native English needs to be further improved. If the language of the article is improved, I think the article is clinically significant and can be published.

Reviewer #3: (No Response)

7. PLOS authors have the option to publish the peer review history of their article (what does this mean?). If published, this will include your full peer review and any attached files.

Reviewer #1: No

Reviewer #2: **Yes: **Yongtao Sun

Reviewer #3: No

---

## [Author Response · Author response to Decision Letter 2]

3 May 2023

Reference: PONE-D-22-33531

Title: Intraoperative intravenous low-dose esketamine improves quality of early recovery after laparoscopic radical resection of colorectal cancer: A prospective, randomized controlled trial

Journal title: PLOS ONE

Authors: Ying Xu, Long He, Shaoxuan Liu, Chaofan Zhang, Yanqiu Ai

Dear Editors:

 Thank you for your useful comments on our manuscript. We wish to give a sincere gratitude to referees for reviewing our paper carefully. We apologize for any inconveniences caused by these errors. We have modified the manuscript accordingly, and the specific modifications are listed point by point below. The responses to the reviewers’ comments are marked using yellow highlight and presented following.

 We would love to thank you for allowing us to resubmit a revised copy of the manuscript and we highly appreciate your time and consideration.

Best regards

Sincerely yours

Journal Requirements:

Response: Thank you for the suggestion. In the reference list, the reference 5 was changed as suggested by reviewer 1 “In the first paragraph, you need to explain the operative related complication that could be improved by the study drug. Otherwise, it is better to remove the sentence starting with “However, the establishment…” as the role of the study drug is not to improve or impact those complications. You may consider other complications, such as inadequate pain management and use of opioids, which could be affected by the study drug.” during major revision. And, the reference 10 was added as suggested by reviewer 1 “In the second paragraph, mention some of the adverse effect of ketamine in comparison to the esketamine.” during major revision.

Thanks again for your advice and I hope to learn more from you.

Additional Editor Comments:

Thank you for your nice revision. There is a very minor statistical comment which needs correction before final acceptance of the manuscript. Well done.

Response: We are very grateful to your comments for the manuscript. We have made a correction in the statistical analysis section.

Thanks again for your advice and I hope to learn more from you.

Reviewer: 1

please state that "Adjustment for multiple testing was done using Bonferroni or LSD" in the statistical analysis section

Response: Thank you for the suggestion. Your suggestion are very important and have important guiding significance for my thesis writing and scientific research work. We have stated that “Adjustment for multiple testing was done using Bonferroni or LSD" in the statistical analysis section.

Thanks again for your advice and I hope to learn more from you.

Reviewer: 2

Thanks to the authors of the manuscript for their active and effective revisions. The quality of the manuscript has been greatly improved, basically meeting the requirements for publication. After my own careful examination, i feel that the native English needs to be further improved. If the language of the article is improved, I think the article is clinically significant and can be published.

Response: Thank you for the suggestion. I have consulted the teachers who is better at English around me to help make the revision, including some inaccuracies in the wording.

Thanks again for your advice and I hope to learn more from you.

---

## [Editor Report · Decision Letter 3]

19 May 2023

Intraoperative intravenous low-dose esketamine improves quality of early recovery after laparoscopic radical resection of colorectal cancer: A prospective, randomized controlled trial

PONE-D-22-33531R3

Dear Dr. Ai,

We’re pleased to inform you that your manuscript has been judged scientifically suitable for publication and will be formally accepted for publication once it meets all outstanding technical requirements.

Kind regards,

Ahmed Mohamed Ibrahim Hasanin

Academic Editor

PLOS ONE

---

## [Editor Report · Acceptance letter]

24 May 2023

PONE-D-22-33531R3 

Intraoperative intravenous low-dose esketamine improves quality of early recovery after laparoscopic radical resection of colorectal cancer: A prospective, randomized controlled trial 

Dear Dr. Ai:

I'm pleased to inform you that your manuscript has been deemed suitable for publication in PLOS ONE. Congratulations! Your manuscript is now with our production department. 

Kind regards, 

on behalf of

Dr. Ahmed Mohamed Ibrahim Hasanin 

Academic Editor

PLOS ONE